# Leaching of Silver and Gold Contained in a Sedimentary Ore, Using Sodium Thiosulfate; A Preliminary Kinetic Study

Edmundo Roldán-Contreras [1], Eleazar Salinas-Rodríguez [1,*], Juan Hernández-Ávila [1], Eduardo Cerecedo-Sáenz [1,*], Ventura Rodríguez-Lugo [1], Ricardo I. Jeldres [2] and Norman Toro [3,4]

[1] Área Académica de Ciencias de la Tierra y Materiales, Universidad Autónoma del Estado de Hidalgo, Carretera Pachuca-Tulancingo km 4.5, Mineral de la Reforma, Hidalgo 42184, Mexico; ed.roldan@hotmail.com (E.R.-C.); herjuan@uaeh.edu.mx (J.H.-Á.); ventura.rl65@gmail.com (V.R.-L.)
[2] Departamento de Ingeniería Química y Procesos de Minerales, Facultad de Ingeniería, Universidad de Antofagasta, Antofagasta 1240000, Chile; ricardo.jeldres@uantof.cl
[3] Departamento de Ingeniería Metalúrgica y Minas, Universidad Católica del Norte, Antofagasta 1270709, Chile; ntoro@ucn.cl
[4] Departament of Mining, Geological and Cartographic Department, Universidad Politécnica de Cartagena, 30202 Murcia, Spain
* Correspondence: salinasr@uaeh.edu.mx (E.S.-R.); mardenjazz@yahoo.com.mx (E.C.-S.); Tel.: +52-771-207-4171 (E.S.-R.)

**Abstract:** Some sedimentary minerals have attractive contents of gold and silver, like a sedimentary exhalative ore available in the eastern of Hidalgo in Mexico. The gold and silver contained represent an interesting opportunity for processing by non-toxic and aggressive leaching reagents like thiosulfate. The preliminary kinetic study indicated that the leaching process was poorly affected by temperature and thiosulfate concentration. The reaction order was $-0.61$ for Ag, considering a thiosulfate concentration between 200–500 mol·m$^{-3}$, while, for Au, it was $-0.09$ for a concentration range between 32–320 mol·m$^{-3}$. By varying the pH 7–10, it was found that the reaction order was n = 5.03 for Ag, while, for Au, the value was n = 0.94, considering pH 9.5–11. The activation energy obtained during the silver leaching process was 3.15 kJ·mol$^{-1}$ (298–328 K), which was indicative of a diffusive control of the process. On the other hand, during gold leaching, the activation energy obtained was of 36.44 kJ·mol$^{-1}$, which was indicative that this process was mixed controlled process, first at low temperatures by diffusive control (298–313 K) and then by chemical control (318–323 K).

**Keywords:** thiosulfate; gold leaching; silver leaching; kinetic analysis; sedimentary ore; diffusion control; mixed control

## 1. Introduction

The mining activity in Mexico has focused mainly on gold and silver ores, the processing of which entails grinding, froth flotation, and cyanidation stages. While the latter involves low operation costs, the reagent utilized is highly toxic, which has even been outlawed in several countries worldwide [1–3]. Moreover, particular operational challenges may arise, wherein the lasting cyanidation process can require up to 24 h [3]. This impairs the leaching efficiency since refractory minerals can be produced that encapsulate in small pyritic and quartz-type particles [4,5], and some cyanide-consuming minerals may also appear, hampering the suitable extraction of metallic contents [5–7].

To date, several studies have been carried out to find leaching reagents that can replace cyanide, falling into chloride, thiourea, and thiosulfate [8,9]. Chloride is corrosive and can cause hazardous working conditions, together with low selectivity during the extraction process [4,6,8]. Processing with thiourea is expensive, and the background indicated that it is a potential carcinogen reagent [10,11]. However, leaching using thiosulfate is an economical and promising method for ore treatment. It offers high reaction selectivity during the process, reduced environmental risks, and low corrosive solutions, low price, offering also an efficient dissolution medium for refractory ores [12–19].

Several studies have displayed good performance on the use of thiosulfate instead of cyanide to extract gold and silver [20–23]. However, a particular challenge persists that is mostly related to the low stability of thiosulfate ions, where proper alternative considers the solution of copper–ammonium–thiosulfate. The $Cu^{2+}$ ions may oxidize the gold and silver, while the thiosulfate makes quite stable complexes with them, which might allow a suitable extraction from ores and waste. Also, the ammonia ions can also form a stable complex with copper ions, avoiding their precipitation [9]. Some studies have shown their performance in the presence of additives and electrolytes, considering the effect of ligands and oxidants reagents [24–27].

Recent studies have found that during the kinetic of silver leaching using thiosulfate, the overall process is controlled by the mass transfer of oxygen to the solid-liquid interface [28]. Additionally, other similar studies done for silver leaching have concluded that for different ranges of concentrations of ammonia and thiosulfate, silver complexes preferentially with thiosulfate [29–32]. On the other hand, the only company that at present is using the thiosulfate solutions for gold leaching is Barrick Gold Corporation (Elko, NV, USA), after an acidic or alkaline pressure oxidation pretreatment [33,34]. The carbonaceous gold ore cannot be treated with cyanide due to the "pre-robbing" phenomena, which does not occur during the leaching using thiosulfate solutions. The weak affinity of carbonaceous material for the gold thiosulfate complex forwards this stage [35]. Some authors have gotten extractions of 11% for gold and 21% for silver using both ammonium thiosulfate with the addition of $H_2O_2$ and cupric ions [36]. On the other hand, thiosulfate easily can be decomposed by some factors, like the Cu(II) content and the presence of different minerals like pyrite and hematite, promoting also high amounts of thiosulfate consumption during leaching process, generating diverse polythionates and at the end, $S^0$, $S^{2-}$, and $SO_3^{2-}$, which can be deposited on the surface of mineral, passivating the dissolution of the metals of interest [9]. Therefore, this work aimed to analyze the dissolution kinetics of gold and silver from a sedimentary ore, where silver can be presented as metallic and/or sulfur, and gold is joint to carbonaceous material. Air-$Na_2S_2O_3^{2-}$ solutions were used without adding cupric ions, which allowed evaluating how thiosulfate could extract these metallic values. The ore considered in this study might have some trace copper contents (less than 10 ppm, perhaps like chalcopyrite) that acted as an oxidizing reagent, improving the gold and silver leaching. Finally, the mechanisms that control the chemical reactions of both metals were discussed.

## 2. Materials and Methods

### 2.1. Materials

An ore, located at the northeast of the State of Hidalgo, Mexico, was collected selectively [37], taking 50,000 g of each sample in 4 different points of the mineralized zone of the outcrop. Samples collected were mixed and quartered, taking a representative sample to carry out the kinetics leaching study. The mineralogical characterization was executed to get accurate data of the phases present, for which an analysis of general phases was proposed through X-Ray diffraction (XRD). The samples were ground up to get an average particle size, less than 78 μm, and kept in an Equinox 2000 X-ray Diffractometer (INEL, Artenary, France, located at UAEH) with $CoK\alpha_1$ radiation. The identification of present phases was executed using the COD Inorganics 2015 databases, which is included in the crystallography open database match software (v.1.10, Crystal Impact, Bonn, Germany).

The scanning electron microscopy (SEM) identified texture, particle sizes, and morphology of the detected phases. The semi-quantitative and punctual analysis was done by energy dispersive spectrometry of X-ray (EDS) (OXFORD Instruments, Oxford, UK). It determined the punctual and semi-quantitative composition of the previously identified particles, using a JEOL scanning electron microscope JSM-IT300 (JEOL Ltd., Tokyo, Japan, located at UAEH, Apan, Mexico)) and an OXFORD X-ray detector (OXFORD Instruments, Oxford, UK) with 30 kV of acceleration voltage. The analysis of samples was done using powders of the sample placed in uniform layers, where punctual semi-quantitative routines were done on scanning areas about 4.5 mm$^2$.

An inductively coupled plasma spectrometry (ICP-MS) analysis was performed by Actlabs (Activation Laboratories Ltd., Ancaster, ON, Canada); this determined the average total rock composition of the mineralized phase, where the positive anomalies of the light rare earth and mineral contents of the platinum group (PGE) were found. In this case, the samples were fused and then diluted and analyzed by a Perkin Elmer Sciex ELAN 9000 ICP-MS spectrometer (Located at Actlabs, Ancaster, ON, Canada). Fused blank was run in triplicate every 10 samples, and then the instrument was recalibrated after every 44 samples.

An analysis by copelation determined the Au and Ag contents. This was executed in an oven EMISON brand, CL Series (Located at UAEH, Pachuca, Mexico). The sample preparation was done utilizing borax, PbO, bone ash, sodium carbonate as flux. The melting temperature was of 1273 K (1000 °C) during 90 min. In the slag separation, a button contained the values of Ag and Au. The release of these metals was executed in porcelain crucibles on a heating plate at 313 K (40 °C), adding 15% nitric acid to obtain a solution of silver nitrate. Then, the release of Au and Ag was done using regal water, adding 10% of hydrochloric acid in test tubes. The determination of Au and Ag contents was done using an ICP-OES Varian brand 735ES ICP (Located at UAEH, Pachuca, Mexico), where samples were analyzed with a minimum of 10 certified reference materials, all prepared with sodium peroxide fusion. Every 10 samples were analyzed by duplicate, and the blank was renewed after every 30 samples measurement. For the kinetics study, the samples were ground during 360 s at a working speed of 150 s$^{-1}$ with a ball charge of 10,230 g, a pulp charge of 2100 g, and a volume of $9.5 \times 10^{-4}$ m$^3$ of water in the ball mill "Denver" (Located at the UAEH-Mexico). Then, wet sieve determined the particle size distribution. Finally, the samples were separated for the next stage of kinetics leaching.

## 2.2. Experimental Procedure

The experiments for the kinetics study were executed in a 0.001 m$^3$ flat bottom glass reactor mounted on a hot plate having a magnetic stirring system and coupled to a pH meter. The pH was monitored and adjusted continuously with NaOH solution at a concentration of 200 mol·m$^{-3}$. A thermocouple attached to the hot plate controlled the temperature. All assays were done at an open atmosphere with vigorous mechanical stirring (500 s$^{-1}$). The chemical reagents were of analytical grade, and distilled water was used for the preparation of solutions. Thus, leaching reagent was added like sodium thiosulfate from Sigma MEYER brand with an essay quality of [Na$_2$S$_2$O$_3$·5H$_2$O] of 99.5–100%.

For the leaching of silver, the kinetics leaching s using thiosulfate solutions was done using the following experimental conditions: concentration of sodium thiosulfate [Na$_2$S$_2$O$_3$], 200 to 500 mol·m$^{-3}$; temperature range, 298 to 328 K; mineral weight, 40 g·m$^3$; pH range, 7 to 10; volume of reaction, 0.0005 m$^3$; stirring rate, 500 s$^{-1}$. This allowed comparing with previous results on mining waste [38]. For the leaching of gold, the kinetics study was executed under the following conditions: concentration of [Na$_2$S$_2$O$_3$], 32 to 320 mol·m$^{-3}$; temperature range, 298 to 323 K; mineral weight, 40 g·m$^{-3}$, pH range, 9.5 to 11; volume of reaction, 0.0005 m$^3$; stirring rate, 500 s$^{-1}$. This followed recommendations stated previously [39,40].

The progress of all experiences was followed by sampling at pre-set times (0–14,400 s) throughout the experiment, and then the dissolved Au and/or Ag were analyzed by ICP and AAS. Mathematical calculations corrected variations in the mass balance in the sampling addition of a reagent.

## 3. Results

### 3.1. Mineral Characterization

The XRD analysis (Figure 1) showed that the mineral species involved were characteristics of a sedimentary exhalative ore [37]. This was principally composed of quartz, ilmenite, and monazite, with some contents of precious metals, such as Pt, Pd, Au, and Ag. Some light rare earths could be present (Table 1), and also trace contents of base metals sulfurs of Pb, Zn, Cu, chalcopyrite, and pyrite, associated with organic material (carbonaceous substance) with attractive contents of gold (5 g/ton) and silver (25 g/ton), determined by ICP, AAS, and cupellation test (Figure 2). All the above gave additional value to this ore.

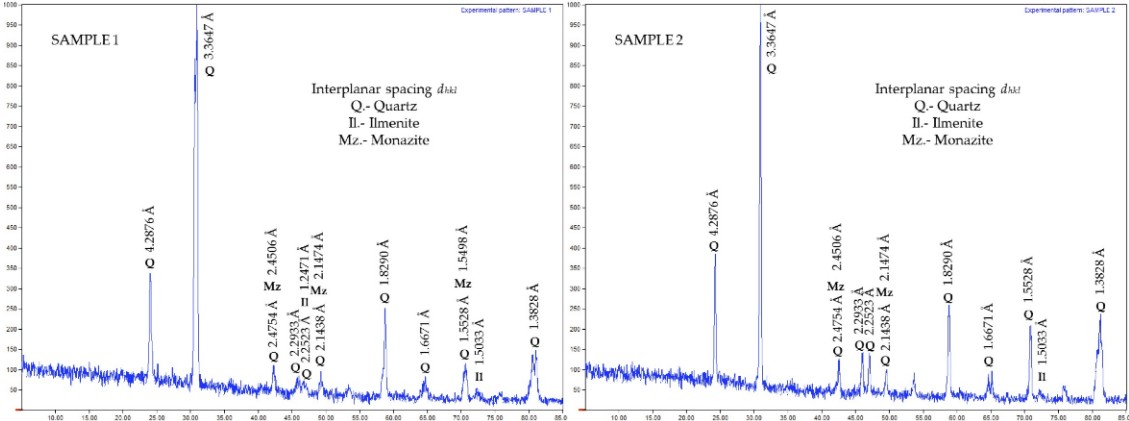

**Figure 1.** XRD spectra of sedimentary ore, used for kinetic study of gold and silver leaching.

**Table 1.** Average chemical composition of mineral executed by ICP-OES/MS, XRF, SEM-EDS, and AAS.

| Element | % wt |
| --- | --- |
| S | 34.3 |
| Fe | 32.6 |
| O | 16.6 |
| C | 2.4 |
| Si | 0.8 |
| Mg | 0.9 |
| Al | 0.6 |
| Na | 0.6 |
| K | 0.05 |
| Ti | 0.006 |
| Cu | 0.0007 |
| Pt | 0.0002 |
| Pd | 0.0007 |
| Au | 0.0005 |
| Ag | 0.0025 |

The morphology obtained by SEM-SE showed irregular shape particles, typical of the mineral species mentioned above, that presented irregular faces in similar dimensional style, having sizes that vary from 30 to 354 μm. Figure 3 shows a general image of ore particles used in this study, also showing the size distribution of particles of this material and the anhedral type of morphology. A detailed zone from where an EDS was executed, revealing the presence of Au and Ag, is also shown in Figure 3. The particle size distribution is shown in Table 2, where the predominant particle (77.6%) had 44 μm of diameter.

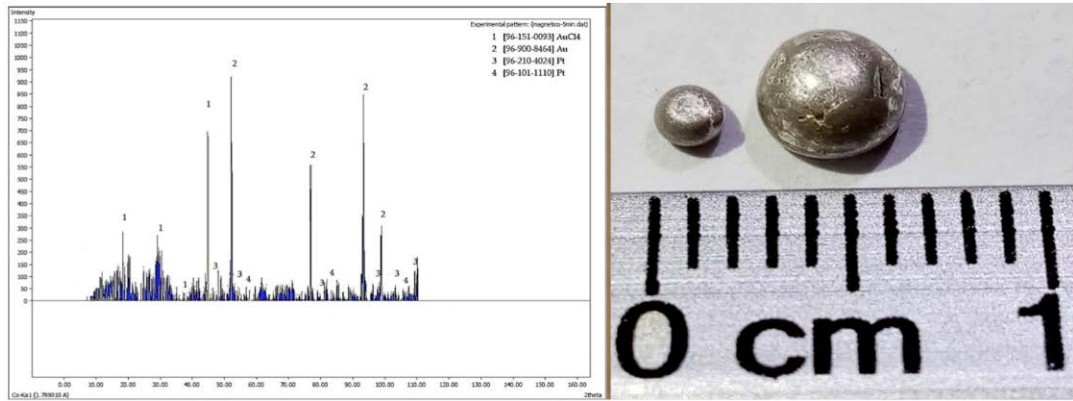

**Figure 2.** Image of the button obtained by copelation and the corresponding XRD spectra.

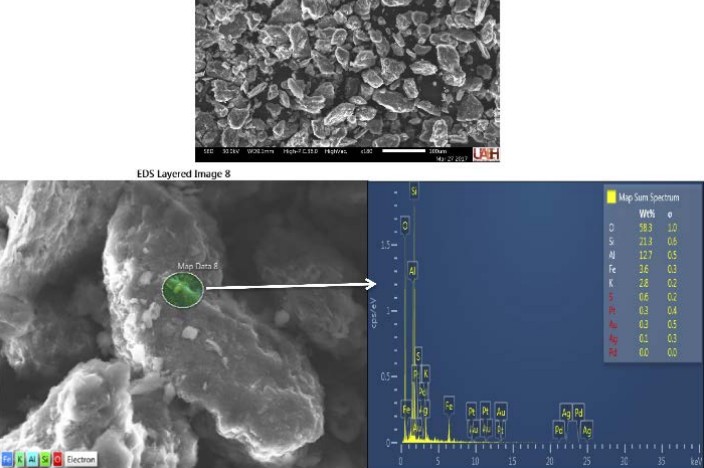

**Figure 3.** The general image of the ore particle size distribution, and detailed zone with an SEM-EDS analysis (SEM-SE), similar for all samples used in this study.

**Table 2.** Ore particle size distribution.

| D (μm) | Partial Weight (pct) | Accumulative Weight (pct) |
|--------|---------------------|---------------------------|
| 37 | 3.8 | 3.8 |
| 44 | 77.6 | 81.4 |
| 53 | 8.5 | 89.9 |
| 63 | 3.5 | 93.4 |
| 105 | 2.7 | 96.1 |
| 125 | 2.2 | 98.3 |
| 149 | 1.5 | 99.8 |
| 354 | 0.2 | 100 |

### 3.2. Kinetic Study of Gold and Silver Leaching

#### 3.2.1. Nature and Stoichiometry of Reactions

The experimental conditions for silver leaching were: concentration of $[Na_2S_2O_3]$, 500 mol·m$^{-3}$; temperature, 298 K; pH, 9; mineral weight, 40 g·m$^{-3}$; stirring rate 500 s$^{-1}$; the volume of reaction, 0.0005 m$^3$, reaching a maximum silver recovery of 80%. For gold leaching, the experimental conditions were the following: concentration of $[Na_2S_2O_3]$, 130 mol·m$^{-3}$; temperature, 298 K; pH, 9.5; mineral

weight, 40 g·m$^{-3}$; stirring rate 500 s$^{-1}$; the volume of reaction, 0.0005 m$^3$, with a maximum recovery of 20%. Figure 4 shows a representation of the leached element (Ag or Au) versus time. For silver, there was no induction period, and the reaction started immediately, describing the progressive conversion until the end of the reaction, where it appeared a stabilization zone (Figure 4A). For gold leaching (Figure 4B), the graph showed a curve type "S" with a small period of induction, then a period of progressive conversion, a stabilization zone. The existence of a short induction period could be caused by the presence of pyrite, which could influence thiosulfate decomposition, leading to a slow gold dissolution.

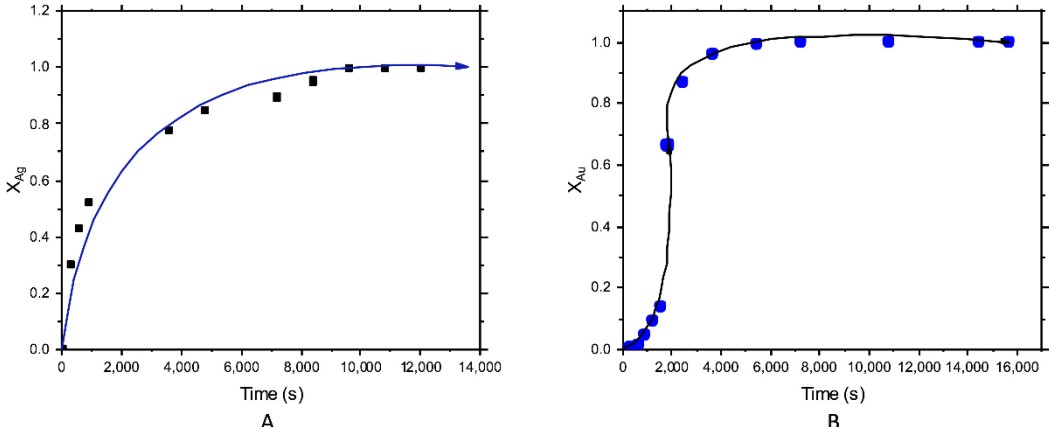

**Figure 4.** Graph of the curve type "S", showing the period of progressive conversion for; (**A**) silver leaching and (**B**) gold leaching, in thiosulfate solutions.

The results showed in Figure 4 were managed according to the core model for diffusive (Equation (1)) and chemical control (Equation (3)) [41–43], determining which kinetics leaching model fit better.

$$[1 - \frac{2}{3} X - (1-X)^{\frac{2}{3}} ] = k_{exp} \times t \tag{1}$$

where

$$k_{exp} = \frac{2 V_M\, D_c\, c_A}{r_0^2} \tag{2}$$

$$[1 - (1-X)^{\frac{1}{3}} ] = k_{exp} \times t \tag{3}$$

where

$$k_{exp} = \frac{V_M\, k_q\, c_A^n}{r_0} \tag{4}$$

$X$ is the reacted fraction of Ag or Au, $V_M$ is the molar volume of the mineral, $c_A$ is the concentration of leaching reactant (in this case thiosulfate), $D_e$ is the diffusion coefficient through the product layer, $k_q$ is the kinetic coefficient, $r_0$ is the initial radius of particle (in average), $k_{exp}$ is the experimental constant, and, finally, $n$ is the order of reaction.

Figure 5 shows the results for silver leaching, considering the fit of experimental data to (A) core model for diffusive control and (B) core model for chemical control. A better representation of the diffusive control model could be appreciated. The same behavior occurred for gold leaching in the thiosulfate solution, and then the kinetics study was analyzed according to the core model for diffusive control.

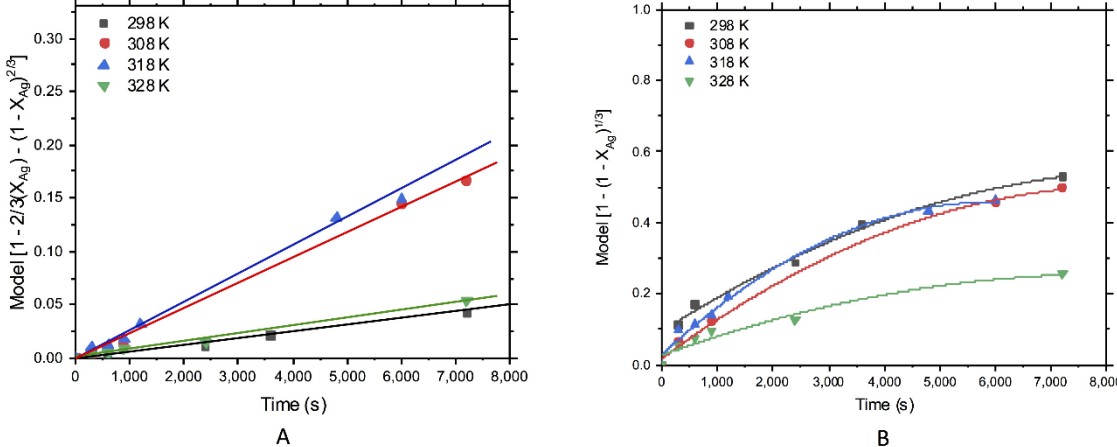

**Figure 5.** Treatment of silver leached data with kinetic models; (**A**) diffusive control and (**B**) chemical control.

### 3.2.2. Stoichiometry of Leaching of Gold and Silver

Mineralogical complexities prevented the determination of stoichiometry for gold and silver leaching system under study. However, a theoretical estimation based on the characterization results determined the presence of gold and silver in the sedimentary ore. These appeared in native form for both metals and like sulfur for the case of silver.

For the case of metallic gold:

$$4Au(s) + 8(S_2O_3)aq^{-2} + O_2g + 2H_2Oaq \rightarrow 4Au(S_2O_3)aq^{-2} + 4OH(aq)^- \tag{5}$$

For the metallic silver:

$$2Ag_{(s)} + 4(S_2O_3)^{-2}_{(aq)} + O_{2(g)} + 2H_2O_{(aq)} \rightarrow 2Ag(S_2O_3)^{-3}_{(aq)} + 4OH^-_{(aq)} \tag{6}$$

When silver is like silver sulfide:

$$Ag_2S_{(s)} + 4(S_2O_3)^{-2}_{(aq)} + O_{2(g)} + 2H_2O_{(aq)} \rightarrow 2Ag(S_2O_3)^{-3}_{(aq)} + S^{+2}_{(s)}4OH^-_{(aq)} \tag{7}$$

### 3.2.3. Effect of the Concentration of [$Na_2S_2O_3$]

The study of gold and silver extraction contained in a sedimentary ore, using the $Na_2S_2O_3$-Air system, was done to establish the effect of the [$Na_2S_2O_3$] concentration, temperature, and pH.

Figure 6A shows the leached fraction of Ag that was analyzed by the core model for diffusive control [41–43], which was [$1 - 2/3X_{Ag} - (1 - X_{Ag})^{2/3}$]. Straight lines were obtained, and their slopes represented the experimental rate constant ($k_{exp}$). Thiosulfate concentration had no effect on the rate of reaction. Since all experiences were done at a high stirring rate, the oxygen input could be enough to maintain a stoichiometric excess during the progress of the reaction. This caused a limited effect on the leaching of silver with respect to thiosulfate concentration, getting a low order of the reaction, n = −0.61 (Figure 6B).

Figure 7A shows the leached fraction of Au, which was evaluated by the same core model for diffusive control [$1 - 2/3X_{Au} - (1 - X_{Au})^{2/3}$]. Similar to silver leaching, thiosulfate concentration did not affect the leaching reaction rate. In this case, the presence of pyrite and $Ag_2S$ could promote the generation of $S^0_{(s))}$ and/or $S^{+2}_{(s)}$, which would be deposited on the gold surface, passivating its dissolution. This also provided a low order of reaction, n = −0.09 (Figure 7B).

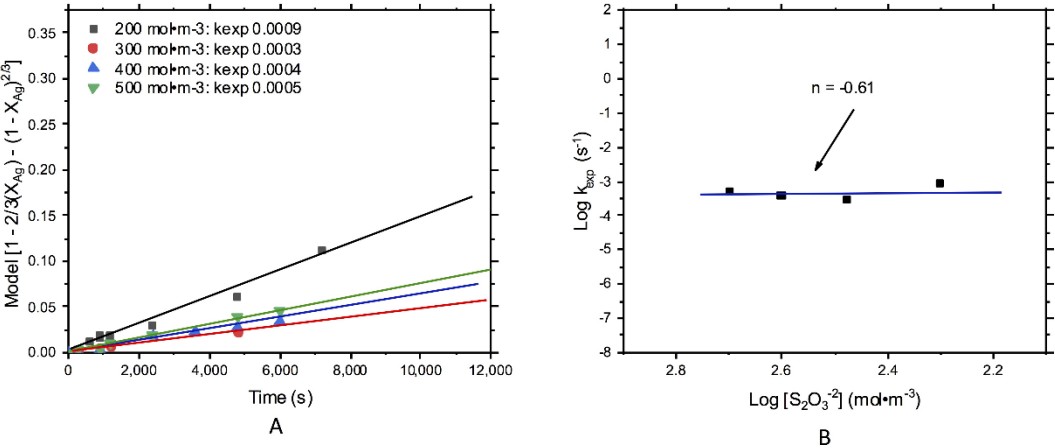

**Figure 6.** Kinetic study of silver leaching; effect of the thiosulfate concentration: (**A**) $k_{exp}$ and (**B**) order of reaction n = −0.61.

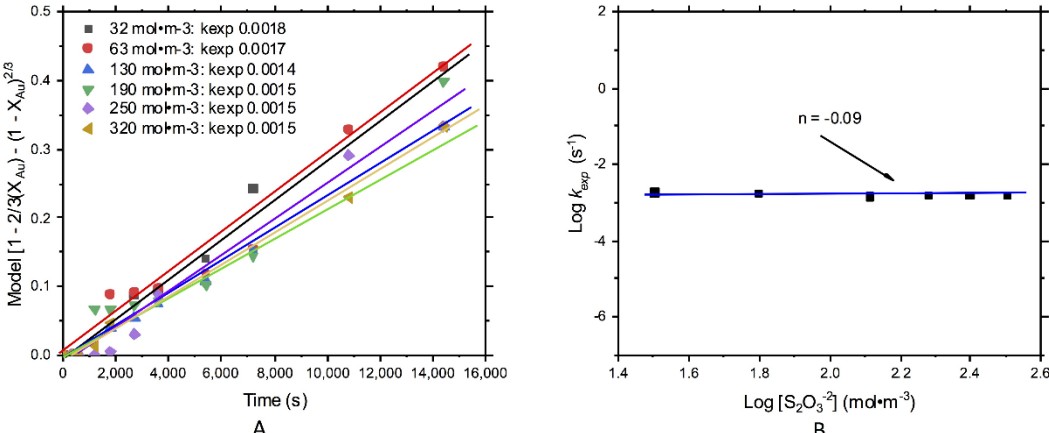

**Figure 7.** Kinetic study of gold leaching; effect of the thiosulfate concentration: (**A**) $k_{exp}$ and (**B**) order of reaction n = −0.09.

### 3.2.4. Effect of the Temperature

The effect of temperature for silver leaching in thiosulfate solutions is shown in Figure 8A,B. For low and high temperatures (298–328 K), the values were quite similar, giving low reaction rates. In conclusion, the influence of temperature was limited.

The activation energy was calculated by plotting the natural logarithm of $k_{exp}$ against the reciprocal of temperature. The slope (m= −($E_a$/R)) of the linear curve represented the activation energy ($E_a$) divided by the negative value of the universal gas constant (Figure 8B). For silver leaching, the calculated energy of activation was of 3.15 kJ/mol, which was representative of a diffusive control [41–43]. This was also observed by the fitting obtained with the diffusive control model. The obtained order of reaction was low, and the curve type "S" had no induction period, then the process was not dependent on both thiosulfate and temperature, and the slow diffusion of products from the particle's surfaces to the deep of solution controlled the overall process.

For gold leaching, the effect of temperature according to the diffusive control model is shown in Figure 9A. The rate constants were quite similar in the range of temperature from 298 to 313 K (0.0011–0.0014), but he slight increase at higher temperatures (318 and 323 K) could indicate that the process was carried out by a combination of both chemical and diffusive control. Besides, the activation energy that is shown in Figure 9B was 36.44 kJ/mol, which, according to the literature, is representative of mixed control [41–43].

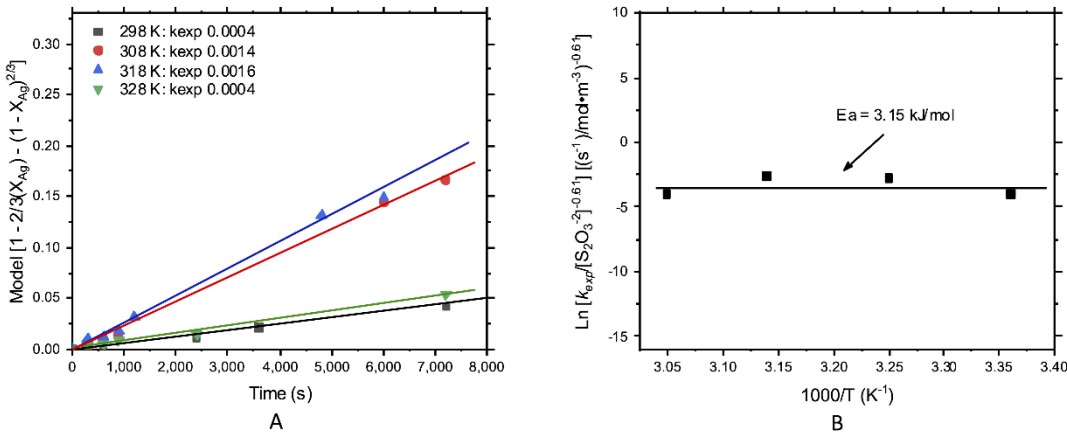

**Figure 8.** Kinetic study of silver leaching; effect of the temperature: (**A**) $k_{exp}$ and (**B**) energy of activation, Ea = 3.15 kJ/mol.

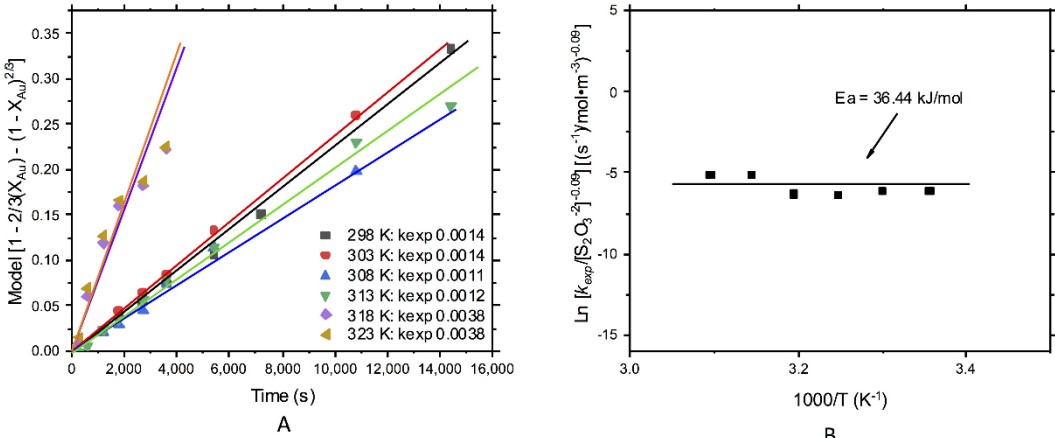

**Figure 9.** Kinetic study of gold leaching; effect of the temperature: (**A**) $k_{exp}$ and (**B**) energy of activation, Ea = 36.44 kJ/mol.

According to the obtained results, the kinetic expressions for silver (Equation (7)) and gold (Equation (8)) leaching in $S_2O_3^{-2}$ medium were:

For silver, diffusive control

$$\frac{r_0^2}{V_M} \left[1 - \frac{2}{3} X_{Ag} - \left(1 - X_{Ag}\right)^{\frac{2}{3}}\right] = 2D_e \left[S_2O_3^{-2}\right] \times t \tag{8}$$

For gold, when control could be by chemical reaction

$$\frac{r_0^2}{V_M} \left[1 - \left(1 - X_{Au}\right)^{\frac{1}{3}}\right] = 3.736 \, x10^3 \, exp^{\frac{-36,440}{RT}} \left[S_2O_3^{-2}\right]^{-0.09} \times t \tag{9}$$

For gold, when control could be by diffusion of products through the product layer

$$\frac{r_0}{V_M} \left[1 - \frac{2}{3} X_{Au} - \left(1 - X_{Au}\right)^{\frac{2}{3}}\right] = 2D_e \left[S_2O_3^{-2}\right] \times t \tag{10}$$

where $V_M = 1.44 \times 10^{-9}$ m$^3 \cdot$mol$^{-1}$ for silver, and $V_M = 6.22 \times 10^{-9}$ m$^3 \cdot$mol$^{-1}$ for gold, $R = 8.31$ J$\cdot$mol$^{-1} \cdot$K$^{-1}$, $r_0$ in m, $D_e$ is the diffusion coefficient through the product layer, T in Kelvin, $[S_2O_3^{-2}]$ is in mol$\cdot$m$^{-3}$, and $t$ is in seconds.

### 3.2.5. Effect of the pH

The effect of the pH for silver leaching is shown in Figure 10A,B. Figure 10A represents the treatment of results with the diffusive control model and the obtained experimental rates. The rates decreased with increasing the pH, which was faster at pH 7, having a poor effect at a higher pH. Figure 10B shows that the influence of pH over the overall reaction rate was significant, finding an order of reaction of n = 5.03.

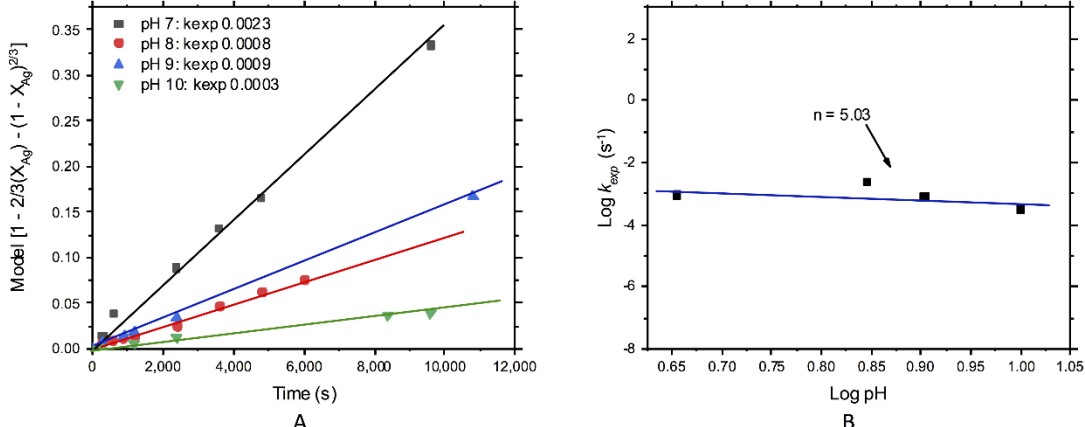

**Figure 10.** Kinetic study of silver leaching; effect of the pH: (**A**) $k_{exp}$ and (**B**) order of reaction n = 5.03.

To conclude, for gold leaching, the pH effect is shown in Figure 11A,B. Unlike what happened with the leaching of silver, the experimental rate constants were similar for all the analyzed pH values. Consequently, a poor effect of this variable over the overall rate of the process was observed (Figure 11A). However, all experimental rate constants were low. This might be explained by the presence of small amounts of sulfur minerals since their formation is inevitable even in small quantities, promoting a partial degradation of thiosulfate. The deposition of this formed sulfur on the surface of the gold particles promoted the decrease in its dissolution rate. Finally, Figure 11B displays the effect of pH over the overall gold leaching reaction rate. The order of reaction was n = 0.94, concluding that the pH had no effect on gold leaching under the thiosulfate concentration and temperatures considered.

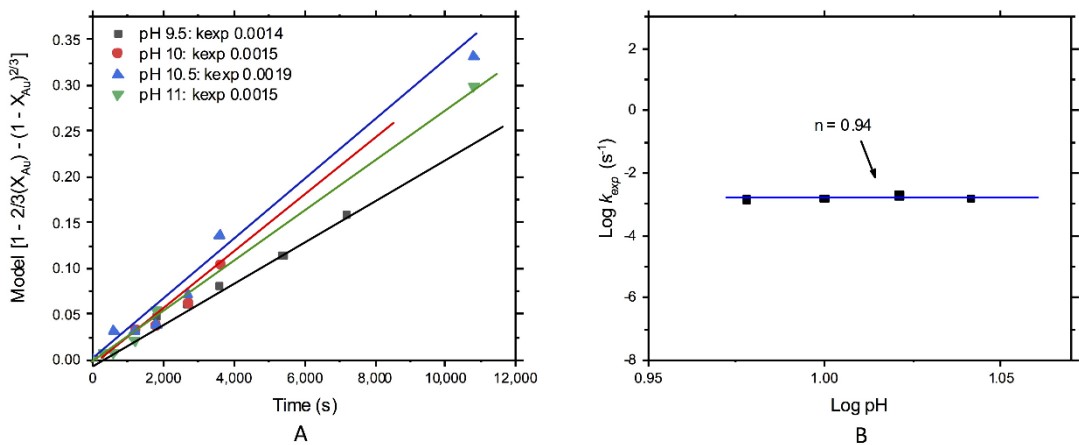

**Figure 11.** Kinetic study of gold leaching; effect of the pH: (**A**) $k_{exp}$ and (**B**) order of reaction n = 0.94.

## 4. Discussion

The mineral studied here presented significant amounts of Ag and Au, including light rare earths contents, which increased its commercial importance. However, this research aimed to analyze the leaching kinetics of gold and silver in thiosulfate solutions without adding Cu(II), which

allowed verifying the ability of the leaching reagent according to the nature and composition of the ore. Although leaching using thiosulfate solutions is considered a viable alternative to cyanidation [16–18,39], nowadays, it is not used at a big scale due to the high reagent consumption and the difficulty in recovering metallic gold and silver [9]. Still, it works appropriately with metallic ores containing carbonaceous materials [35], like the ore here studied (having about a 2% C).

The use of thiosulfate as a leaching reagent needs more attention since some problems can arise, the reagent is unstable, and it can self-decomposed or reduced to $S^0$, $S^-$, and $S_3^{-2}$. These species may deposit on the surface of the metal, hindering its dissolution [25]. Additionally, the reaction is prolonged without the addition of Cu(II) and ammonia [44]. However, for this study, we decided to use sodium thiosulfate solely since the characteristics of mineral gave a chance to get promising results. Yen et al. [45] concluded that high concentrations of thiosulfate, high dissolved oxygen, and high temperatures increased the consumption of thiosulfate. Consequently, low metallic recoveries were obtained. In parallel, dilute concentrations of thiosulfate, low oxygen concentration, and low temperatures could reduce the rate of gold dissolution. However, the results of this study were encouraging, getting 80% of silver recovery using 500 mol·m$^3$ of sodium thiosulfate, pH 7, the temperature of 298 K, stirring rate of 500 s$^{-1}$, and using air in low concentration (just by incorporation during mechanical stirring). In the case of gold, the maximum dissolution was of 20% using 130 mol·m$^3$ of sodium thiosulfate, pH 9, the temperature of 298 K, stirring rate of 500 s$^{-1}$, and using air in low concentration (just by incorporation during mechanical stirring). As known, the experimental conditions constantly change during leaching, and it could be difficult to control each of them with adequate precision. Similarly, the use of relatively low leaching reagent concentrations with limited oxygen supply is an easy method to avoid high thiosulfate consumption during the leaching process [9].

Some authors [28,38] reported interesting results for the silver leaching with thiosulfate solutions, concluding that the diffusion of oxygen controls the process through the product layer. The order of the reaction was similar to that found here, which was n = −0.61 for thiosulfate concentrations 200–500 mol·m$^{-3}$. Other cases displayed values like n = 0.074 for thiosulfate concentrations 100–500 mol·m$^{-3}$, for silver leaching contained in mining burrows [38], n = 0.41 for thiosulfate concentration 25–200 mol·m$^{-3}$ [28], and n = 0 for thiosulfate concentration 200–600 mol·m$^{-3}$, for leaching of metallic silver [28]. Thiosulfate concentration does not affect the reaction rate for silver leaching that contrast works with different mineralogies [9]. For this reason, more in-depth researches are needed to disclose the behavior of thiosulfate solutions over silver leaching, according to the nature of the minerals and species involved.

For gold leaching, the literature reports that the thiosulfate concentration, joint with some contents of Cu(II) and ammonia, have a detrimental effect on the gold dissolution by the presence of minerals, such as pyrite and other sulfurs [25]. This promotes a self-decomposition of the thiosulfate and, as a consequence, higher consumption of this reactant. In this study, it was found that the thiosulfate concentration had no effect on the reaction rate getting reaction order of n = −0.09 (quite similar to that obtained for silver leaching). The latter means that the absence of important amounts of sulfur minerals might avoid the formation of $S^0$ and $S^{-2}$, which deposit on the gold surface, reducing the dissolution.

According to the evaluation of the effect of the temperature on the rate of silver leaching, the activation energy found here was $E_a$ = 3.15 kJ/mol. This corresponded to a diffusive control, and it validated the model used for the treatment of data [41–43]. In this case, the overall reaction was controlled by oxygen diffusion through the product layer because the chemical reaction of the complexation of Ag is too fast. The above result was consistent with that obtained during the silver leaching contained in a mining waste [38], (even with the absence of Cu(II) where the apparent energy of activation was of $E_a$ = 1.91 kJ/mol.

When evaluating the effect of temperature during the gold leaching, it was found that the apparent energy of activation was $E_a$ = 36.44 kJ/mol. According to previous studies, this corresponded to a mixed control [41–43]. At low temperatures, the experimental rates were low (0.0011–0.0014 s$^{-1}$), and diffusive control was dominant since thiosulfate was more stable. Then, at higher temperatures, the

instability of thiosulfate could lead to its decomposition, generating $S^0$ and $S^{2-}$ that might deposit on the gold surface [9]. This avoided a fast chemical reaction, being this step that controlled the process.

Finally, due to the concentration of $\left[S_2O_3^{-2}\right]$ could be affected by the pH (below 4 and above 12), where thiosulfate degradation or decomposition occurred, with the consequent formation of elemental sulfur (especially below a pH 4) [28,38]. Consequently, this work was executed between the valid range pH, where the thiosulfate could be more stable and was not pH-dependent. For the leaching of silver, the order of reaction was n = 5.03, indicating an apparent effect of this variable but only at pH 7. At this condition, the rate of reaction was higher than at pH 8–10. For gold dissolution, the order of reaction was n = 0.94, indicating that the pH did not alter the reaction rate, which indicated the stability of the reactant. This was because of the operational conditions used in this work and the absence of minerals like pyrite that could promote thiosulfate decomposition.

**Author Contributions:** The conceptualization was done by E.S.-R., J.H.-Á., and E.C.-S.; the methodology was executed by E.R.-C., E.S.-R., and V.R.-L.; validation was done by E.S.-R.; data curation, E.S.-R.; writing—original draft preparation, E.S.-R., E.R.-C., and J.H.-Á.; writing—review and editing, E.S.-R. and R.I.J.; supervision, E.S.-R. and V.R.-L.; funding acquisition, N.T. and R.I.J. All authors have read and agreed to the published version of the manuscript.

**Funding:** This research received no external funding.

**Acknowledgments:** Authors want to thank the CONACyT of the Mexican Government for its support given through the Ph.D. scholarship to the student Edmundo R. C. (CVU 430931). Thanks also go to the Autonomous University of the State of Hidalgo, Mexico, especially to the Academic Group of Advanced Materials from the AACTyM-ICBI; University of Antofagasta, Chile; Polytechnic University of Cartagena, Spain, and Northern Catholic University, Chile.

**Conflicts of Interest:** The authors declare no conflict of interest.

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
