# Peer review of "Leaching of Silver and Gold Contained in a Sedimentary Ore, Using Sodium Thiosulfate; A Preliminary Kinetic Study"

_metals, doi:10.3390/met10020159_

Round 1

Reviewer 1 Report

The manuscript is written in an unintelligible English. Even the basic concepts employed in study study are misused. "Cyaniding" is totally different from "cyanidation". Cyaniding is a case-hardening process, whereas cyanidation is a hydrometallurgical technique for extracting gold from ores. The whole manuscript is full of bad English that makes it incomprehensible. I suggest you give the manuscript to a good English speaker for revision before submitting it to any journal. 

Author Response

Response to reviewer 1

English language and style: (x) Extensive editing of English language and style required

Response: The paper was reviewed to correct mistakes in the language editing

Comments and Suggestions for Authors

The manuscript is written in an unintelligible English. Even the basic concepts employed in study study are misused. "Cyaniding" is totally different from "cyanidation". Cyaniding is a case-hardening process, whereas cyanidation is a hydrometallurgical technique for extracting gold from ores. The whole manuscript is full of bad English that makes it incomprehensible. I suggest you give the manuscript to a good English speaker for revision before submitting it to any journal. 

Response: The document was reviewed trying to improve English language and also, was changed the word cyaniding in all the text.

Authors have also reviewed and modified the part of the explanation of the results and conclusions, trying to be more explicit in their description

Finally authors thanks for your time and willingness to review this work.

Regards

Eleazar Salinas R.

Reviewer 2 Report

The present research work focuses on the leaching of silver and gold from a sedimentary ore, using sodium thiosulfate. The work is interesting, but major revision should be made prior to acceptance

It very is known that the systems based on leaching using thiosulfates are considered to be non-toxic alternatives to the conventional processes such as cyaniding and an efficient alternative dissolution medium for refractory ores.

However, it is also known the low stability of thiosulfate ions.

So, the Authors should discuss in the Introduction Section the following drawbacks of the proposed process:

The thiosulfate decomposition is also catalyzed by Cu(II). This not only results in very high reagent consumption, but also generates significant amounts of polythionates, which may be detrimental to the down-stream gold recovery process The negative influence of copper on gold recovery from pregnant leach liquor. Copper ions are co-precipitated with dissolved gold during cementation using pulverized metal such as zinc and iron, and also are co-deposited on cathode surfaces during electrowinning. [Cu(S2O3)3] that is the predominant form of Cu(I) in thiosulfate solution seriously interferes with the adsorption or extraction of [Au(S2O3)2] The process is not robust, with the leaching conditions having to be controlled very carefully and often ore- specifically.

Other comments regarding experimental part

The XRD analysis (Figure 1) confirmed that the mineral is a sedimentary exhalative one, which principally is composed by quartz, ilmenite and monazite with contents of precious metals, such as Pt, Pd, Au and Ag; some light rare earths

This is not a conclusion from the X-Ray Diffraction Analysis and it should be rephrased.

Average chemical composition of mineral executed by ICP

What the Authors mean with “chemical composition of mineral”? How they did this analysis? With EDS?

If they use EDS analysis, they should present corresponding micrographs (in polished sections) and the EDS charts.

Furthermore, the Authors should present a full chemical analysis of the ore. What is, for example, the Cu concentration?

The morphology obtained by SEM –SE, showed particles of irregular…

I do not understand what the Authors really gained from SEM examination. Only the particle size of the ore…?

This is totally depended on the grinding process and if the Authors would like to present a representative particle size distribution it would be better to use a laser Particle Size Analyzer, such as Malvern or Cilas.

Furthermore, the finesse of the ore is a factor which significantly affect the leaching process and it should be studied..

On the other hand, it would very helpful if the Authors had performed SEM in polished sections of the ore, in order to identify the nature of the Ag and Au phases in the ore.

The Authors in the Introduction Section mentioned that silver can be present as metallic and/or sulfur, and the gold is free but with carbonaceous material.

These results should be proved by SEM analysis (here we see only angular particles of the ore and nothing else).

Also, there are not EDS analysis, although the Authors mentioned that “quantitative analysis was done by Energy Dispersive Spectrometry of Xray”.

The use of thiosulfate like leaching reagent, needs more attention due to its instability …

The Authors should present o full chemical analysis of a real leach liquor...

Author Response

Response to reviewer 2

English language and style: (x) English language and style are fine/minor spell check required

Response: This point was checked according observation done by the reviewer 1.

Comments and Suggestions for Authors:

The present research work focuses on the leaching of silver and gold from a sedimentary ore, using sodium thiosulfate. The work is interesting, but major revision should be made prior to acceptance

Response: Author have checked all points observed and was modified or re written trying to correct what was indicated by the reviewer to improve the text. the authors appreciate the reviewer's time and dedication that will undoubtedly help in the improvement of this work.

It very is known that the systems based on leaching using thiosulfates are considered to be non-toxic alternatives to the conventional processes such as cyaniding and an efficient alternative dissolution medium for refractory ores.

Response: That is correct, but also the behaviour of leaching process using thiosulfate still has many secrets, depending on the mineral, material or waste that is treated to extract the gold and silver values. In this work, we are trying to disclose some effects on gold and silver leaching involved in a sedimentary ore having organic matter like carbon and small contents of pyrite and chalcopyrite.

However, it is also known the low stability of thiosulfate ions.

Response: this was pointed in line 63

So, the Authors should discuss in the Introduction Section the following drawbacks of the proposed process:

The thiosulfate decomposition is also catalyzed by Cu(II). This not only results in very high reagent consumption, but also generates significant amounts of polythionates, which may be detrimental to the down-stream gold recovery process The negative influence of copper on gold recovery from pregnant leach liquor. Copper ions are co-precipitated with dissolved gold during cementation using pulverized metal such as zinc and iron, and also are co-deposited on cathode surfaces during electrowinning. [Cu(S2O3)3] that is the predominant form of Cu(I) in thiosulfate solution seriously interferes with the adsorption or extraction of [Au(S2O3)2] The process is not robust, with the leaching conditions having to be controlled very carefully and often ore- specifically.

Response: This was included in Introduction part.

Other comments regarding experimental part

The XRD analysis (Figure 1) confirmed that the mineral is a sedimentary exhalative one, which principally is composed by quartz, ilmenite and monazite with contents of precious metals, such as Pt, Pd, Au and Ag; some light rare earths

This is not a conclusion from the X-Ray Diffraction Analysis and it should be rephrased.

Response: This point was checked and corrected.

Average chemical composition of mineral executed by ICP

What the Authors mean with “chemical composition of mineral”? How they did this analysis? With EDS?

Response: This analysis was only done to determine some semi quantitative and punctual composition of some particles. The mass chemical analysis was executed by ICP (Inductively Coupled Plasma Spectrometry (ICP-MS)). Lines 123 – 129 & 137 – 141.

If they use EDS analysis, they should present corresponding micrographs (in polished sections) and the EDS charts.

Response: For this study, the most important was to determine the gold and silver contents, as well as the mineral species associated to them to carry out the kinetics study of leaching using thiosulfate solutions.

Furthermore, the Authors should present a full chemical analysis of the ore. What is, for example, the Cu concentration?

Response: The corrected value of Cu was included in table 1 and was determined by ICP-MS.

The morphology obtained by SEM –SE, showed particles of irregular…

I do not understand what the Authors really gained from SEM examination. Only the particle size of the ore…?

This is totally depended on the grinding process and if the Authors would like to present a representative particle size distribution it would be better to use a laser Particle Size Analyzer, such as Malvern or Cilas.

Furthermore, the finesse of the ore is a factor which significantly affect the leaching process and it should be studied..

Response: Indeed, this analysis was only do determine not only the particle size but also the shape of particles, because this part was of importance during the treatment of the kinetics model used to determine the orders of reaction and the activation energies. In the same way, the particle size distribution was executed by wet sieve, where was found that the average particle size is about to 44 mm shown in table 2.

On the other hand, it would very helpful if the Authors had performed SEM in polished sections of the ore, in order to identify the nature of the Ag and Au phases in the ore.

The Authors in the Introduction Section mentioned that silver can be present as metallic and/or sulfur, and the gold is free but with carbonaceous material.

These results should be proved by SEM analysis (here we see only angular particles of the ore and nothing else).

Also, there are not EDS analysis, although the Authors mentioned that “quantitative analysis was done by Energy Dispersive Spectrometry of Xray”.

Response: An image showing a particle with trace contents of Au and Silver was added, including also an EDS analysis of the marked area. Figure 3 modified. Authors done many EDS analysis founding the presence of Au and Ag (in metallic and like sulphur form), but they think unnecessary to include all them, because only with the cupellation essay could be determined the Au and Ag contents for kinetics study.

The use of thiosulfate like leaching reagent, needs more attention due to its instability …

The Authors should present o full chemical analysis of a real leach liquor...

Response: During kinetics leaching using thiosulfate, experiments were performed only to determine the effect of thiosulfate concentration, temperature and pH, according the metallic vales contained in ore, but also evaluating the effect of mineral species such as chalcopyrite and pyrite (even if they are in trace contents), the size and morphology of particles involved. Also this is a preliminary study the next stage will be carry out to disclose another effect like the consumption of leaching reagent, and the stability of it in the leaching liquor.

Finally authors thanks for your time and devotion to review this paper, which could be of great importance for its improvement.

Regards

Eleazar Salinas R.

Round 2

Reviewer 2 Report

The authors have revised their manuscript according to the reviewers’ comments. The revised manuscript is clearly written and organized. The methodology is in line with the state of the art in academic and industrial research. The grammar and sentence structure are fine. In my opinion, it can be accepted in the current form for publishing in "Metals".